# β-hydroxybutyrate as an Anti-Aging Metabolite

**DOI:** 10.3390/nu13103420

**Published:** 2021-09-28

**Authors:** Lian Wang, Peijie Chen, Weihua Xiao

**Affiliations:** Shanghai Frontiers Science Research Base of Exercise and Metabolic Health, Shanghai University of Sport, Shanghai 200438, China; wanglian9369@hotmail.com

**Keywords:** β-hydroxybutyrate, ketone body, metabolism, supplementation, aging, diseases

## Abstract

The ketone bodies, especially β-hydroxybutyrate (β-HB), derive from fatty acid oxidation and alternatively serve as a fuel source for peripheral tissues including the brain, heart, and skeletal muscle. β-HB is currently considered not solely an energy substrate for maintaining metabolic homeostasis but also acts as a signaling molecule of modulating lipolysis, oxidative stress, and neuroprotection. Besides, it serves as an epigenetic regulator in terms of histone methylation, acetylation, β-hydroxybutyrylation to delay various age-related diseases. In addition, studies support endogenous β-HB administration or exogenous supplementation as effective strategies to induce a metabolic state of nutritional ketosis. The purpose of this review article is to provide an overview of β-HB metabolism and its relationship and application in age-related diseases. Future studies are needed to reveal whether β-HB has the potential to serve as adjunctive nutritional therapy for aging.

## 1. Introduction

Ketone bodies (KB) refer to β-hydroxybutyrate (C4H8O3, β-HB; also known as 3-hydroxybutyric acid), acetoacetate (AcAc), and acetone (Figure 1). They are small molecules primarily synthesized in the liver, produced from fatty acid oxidation-derived acetyl-coenzyme A (Ac-CoA), and transported to extrahepatic tissues for terminal oxidation. β-HB, the most abundant KB, is a water-soluble principal reversibly formed from the reduction of AcAc in the mitochondria [1]. It is present as R-/S-β-HB enantiomers (or termed D-/L-β-HB, respectively), with the R-enantiomer being the predominant circulating KB. Nutritional ketosis (NK) is defined as a nutritionally induced metabolic state of plasma β-HB concentration ≥0.5 mM, which can be achieved endogenously through caloric and dietary administration, mainly by ketogenic diet, or exogenously with oral supplementation, including β-HB and its related forms, resulting in a therapeutic effect in several medical conditions [2,3]. Apart from a role as an alternative energy source for extrahepatic tissues such as the brain, heart, or skeletal muscle, β-HB also acts as a signaling mediator involved in many regulations of cellular functions and adaptive responses [4,5]. It as well plays an epigenetic role in metabolic diseases through alterations in gene expression, cell surface receptor activation, as well as histone modifications.

Although the clinical efficacy of KB therapy is gradually widely recognized, there are still considerations about its potential mechanisms which are not fully clarified yet. In the past two decades, an explosive knowledge of the genetic and metabolic factors that affect aging has been witnessed. And a recent study has reported that the supplementation of D-β-HB to C. elegans (Caenorhabditis elegans, a kind of roundworm which is usually used for aging research) could improve an extension of lifespan, demonstrating the role of β-HB as “an anti-aging ketone body” [6]. This review will describe β-HB metabolism, outline the role of β-HB in age-related diseases, as well as summarize the therapeutic application of endogenous and exogenous β-HB. Hence, we discussed the underlying mechanisms of the metabolic effects of β-HB and explored whether β-HB had the potential to serve as adjunctive nutritional therapy for age-related diseases.

## 2. Overview of β-HB Metabolism

### 2.1. Ketone Body Production and Utilization

#### 2.1.1. Ketogenesis in Liver

KB synthesis occurs mainly in the liver mitochondrial matrix which is called hepatic ketogenesis governed by the physiological and biochemical conversions of fat (Figure 2). Ketogenic amino acids may also undertake ketogenesis, but the production of circulating KB is likely less than 5% [7]. In the fasting state, low insulin combining with high cortisol and glucagon promotes adipocytes to release non-esterified fatty acids, which will enter the bloodstream [8]. Perivenous hepatocytes take up fatty acids in the mitochondria [9,10,11,12]. There is a series of reactions sequentially starting from Ac-CoA as well as acetoacetyl CoA (AcAc-CoA), and then transforming to 3-hydroxymethylglutaryl-CoA (HMG-CoA), finally ending with the releasing of AcAc [13]. Some of the AcAc is exported from the hepatocytes, yet the majority is reduced to β-HB catalyzed by 3-hydroxybutyrate dehydrogenase 1 (BDH1) in an NAD+/NADH-coupled equilibrium reaction [14,15].

#### 2.1.2. Ketolysis in Extra-Hepatic Tissues

Circulating KB are transported through the plasma membrane via the monocarboxylate transporter (MCT) proteins (Figure 2). The expression of MCT is tissue-specific, MCT1 is ubiquitously expressed, MCT2 is specifically expressed in the neurons and kidney, and MCT4 is expressed in skeletal muscle, lung, heart, and glia [16,17,18]. Ketolysis occurs in these extrahepatic organs, including the brain, heart, neurons, kidney cortex, and skeletal muscle, using β-HB for terminal oxidation. This is the metabolic response to some definite physiological situations such as limited carbohydrate (CHO) content, post-exercise, or abnormal insulin signaling [19]. In these extrahepatic tissues, the oxidation of β-HB to AcAc is catalyzed by BDH in the mitochondria. Next, in the rate-limiting reaction of ketolysis, AcAc is activated into AcAc-CoA and succinate by succinyl-CoA:3-oxoacid CoA transferase (SCOT), which is also called 3-oxoacid-CoA transferase 1 (OXCT1). This enzyme is not expressed in the liver, and thus hepatic tissues are incapable of utilizing the β-HB they produce [20,21]. Two molecules of Ac-CoA subsequently transform from AcAc-CoA and enter the TCA cycle for energy requirement.

### 2.2. β-HB as an Energy Substrate

Under the circumstance of CHO restriction, prolonged fasting as well as exercise, fatty acids break down to β-HB, which turns out to be a compensatory energy fuel in the brain, heart, kidneys, and muscles (Figure 2) [22,23,24]. Thus, a ketosis-like condition is generated under the circumstance of lower blood glucose or glycogen depletion in which liver particularly provides fatty acid-derived β-HB [25]. Under physiological situations, the blood KB concentration ranges from 0.05 to 0.1 mM in humans, and when rising over 0.5 mM, it achieves a metabolic state of NK. Moreover, its level may reach 5–7 mM during prolonged fasting [26], starvation, caloric restriction (CR), ketogenic diet (KD) [27,28], exercise [29], insulin deficiency, pregnancy, or neonatal period [17], and even reach 10–25 mM in a particular circumstance of diabetic ketoacidosis [30,31]. 300 g of KB are produced daily by the human liver, and ketones could provide approximately 5% of the energy requirements in the fed state, rising to 20% in long-term fasting [32,33].

### 2.3. β-HB as a Signaling Mediator

β-HB is not only defined as a fat-derived energetic substrate for the brain, heart, and skeletal muscles under prolonged fasting and exercise but also acts as a metabolic signal modulating a wide variety of cellular functions (Figure 3) [34,35].

#### 2.3.1. Cell Surface Receptor

β-HB has been described as a signaling molecule as it is a ligand for G protein-coupled receptors (GPCRs) including GPR41 and GPR109A (HM74A in humans, PUMA-G in mice), inducing the activation or inactivation of various signaling pathways relevant to lipid metabolism or cellular growth [36,37,38]. β-HB can modulate body energy expenditure and metabolic homeostasis through the mechanism which GPR41, known as free fatty acid receptor 3 (FFAR3), suppresses sympathetic nervous system activity via the Gβγ-PLCβ-MAPK signaling pathway [37,39,40]. GPR109A, which is regarded as hydroxy-carboxylic acid receptor 2 (HCAR2), is located on multiple cells such as neutrophils, macrophages, adipocytes, mediating anti-lipolytic activity, inflammation, and atherogenesis [41,42]. Thus, β-HB exerts numerous effects on neurodegeneration, inflammatory processes, and dyslipidemia as an endogenous ligand via the GPR109A signaling pathway [43].

#### 2.3.2. Anti-Oxidative Stress

Oxidative stress is commonly defined as a condition in which the reactive oxygen species (ROS) are in excess, and this might be owing to excessive production or impaired elimination [44,45]. β-HB is associated with the attenuation of oxidative stress, due to its inhibition of ROS production, prevention of lipid peroxidation and protein oxidation as well as increasing the levels of antioxidant proteins [46]. β-HB plays an anti-oxidative effect in vivo and in vitro studies, especially in the aspect of neuroprotection. It is reported that in the year 2000, Kashiwaya and his colleagues found that β-HB could protect neurons from oxidative damage [47]. Treating cells with β-HB could reduce the cytosolic NADP+/NADPH ratio and increase reduced glutathione, which is one of the major cellular low molecular weight antioxidants. It was reported that treatment of lipopolysaccharides (LPS) in cells induced a proinflammatory stimulation, and the signaling pathway of nuclear factor-κB (NF-κB) was inhibited by β-HB via translocation and degradation of NF-κB inhibitor α. The proinflammatory response to LPS was alleviated by the management of β-HB to cells due to NF-κB regulating various proinflammatory genes expression, including interleukin (IL)-1β, IL-6, tumor necrosis factor-α, inducible nitric oxide synthase, and cyclooxygenase-2 [48]. β-HB metabolism may reduce the generation of ROS and enhance antioxidant defenses through altering redox couples ratios in cytoplasm and mitochondria, in which ROS is mainly produced [49].

#### 2.3.3. Neuroprotection

β-HB could inhibit abnormal glycolysis and reduce glucose uptake through altering metabolic activity in neurodegeneration [50]. β-HB plays a role in neuroprotection by ATP production and regulates NADP+/NADPH ratio, glutathione activity, mitochondrial permeability, glycolytic flux, apoptosis, and neuroinflammation, triggering normal synaptic neurotransmission. β-HB supplementation and CR also mitigate oxidative stress, neuronal apoptosis, neuroinflammation, and intensify neurotrophins generation, including brain-derived neurotrophic factor (BDNF), neurotrophin-3 as well as glial cell line-derived neurotrophic factor, which associates with altered neuronal functions, especially in the senescent brain. It has been demonstrated that β-HB not only enhances neuronal respiration, ATP generation, oxidative functions, ATP-sensitive potassium channel activity, but also improves dopamine volume in the mesencephalon, and regulates motor functions as well [51,52].

### 2.4. β-HB as an Epigenetic Regulator

Epigenetics is the study of inherited changes in the gene function or phenotype caused by molecular mechanisms other than changes in DNA sequence [53]. The three major epigenetic mechanisms include DNA methylation, histone post-translational modifications (PTM), and microRNAs (miRNAs) transcriptional regulation. PTM consist of histone methylation, acetylation, phosphorylation, and ubiquitination. Of note, β-HB is particularly regarded as an endogenous inhibitor of class I histone deacetylase (HDAC) as an epigenetic regulator [54]. Besides, it has recently been shown that high levels of β-HB could induce lysine β-hydroxybutyrylation (Kbhb) in histones of cells [55]. β-HB has been shown to play various signaling roles of linking gene transcription and cellular function with the outside environment in epigenetic modifications which could explain its pleiotropic effects on aging (Figure 4).

#### 2.4.1. Histone Deacetylases Inhibition

β-HB has been demonstrated to naturally inhibit HDACs in a dose-dependent manner, including HDAC 1, 3, and 4, which increases histone acetylation, decreases α-synuclein toxicity and prevents dopaminergic neurons from cell death [56,57]. β-HB may regulate HDACs and alter gene expression through these underlying mechanisms. For instance, β-HB-mediated inhibition of HDACs in mice raises BDNF expression, reduces the NAD+/NADH ratio, and increases ATP, which is closely related to adult neurogenesis [58,59]. In rodent models, BDNF also protects the central nervous system from NF-κB-mediated neuroinflammation and apoptosis [60,61]. Moreover, in the basal ganglia of Parkinson’s disease (PD) rats, the activity of various antioxidant enzymes increased and oxidative damage was restrained [62,63]. Therefore, β-HB may protect against energy depletion, oxidative stress, inflammation, autophagy, and apoptosis by inducing BDNF expression through the inhibition of HDACs.

#### 2.4.2. Histone Lysine β-hydroxybutyrylation

Kbhb is recently considered as an adaptation to changes in cellular energy levels, promoting the transcription level of circadian rhythms and energy stress-response genes [64]. Kbhb as a newly identified epigenetic mark integrates the classic DNA methylation with PTM, offering a novel way to explore chromatin regulation and various functions of β-HB under the circumstance of important human pathophysiological states [65,66,67]. It has been reported that β-HB-mediated p53 Kbhb, which is an essential tumor suppression factor, is dramatically increased in cultured cells treated with β-HB, explaining the association between Kbhb and tumor as well as providing a promising therapeutic target for cancer treatment [68]. A recent study has demonstrated that the p300-dependent Kbhb pathway could directly mediate in vitro transcription through catalyzing the enzymatic addition, representing an adapted epigenetic mechanism for various cellular processes and physiopathological conditions [69]. Consequently, the actions of Kbhb may play vital roles in the pathogenesis and treatment of diverse human diseases.

## 3. Role of β-HB in Age-Related Diseases

### 3.1. Relationship between β-HB and Aging

It has been demonstrated that CR plays an important role in prolonging life and delaying the onset of age-related diseases in multiple species, such as mice, rats, fish, worms, flies, and yeast [70]. KD is considered to promote a longer lifespan the same as CR [71]. Circulating β-HB is seen as an antiaging metabolite, as it is increased significantly during CR and KD [72]. Moreover, the supplementation of β-HB could extend the lifespan of C. elegans and regulate aging and longevity [73]. β-HB diminishes senescence-associated secretory phenotype as well as senescent vascular cells in mammals [74]. Although a variety of studies verify the therapeutic effects of KD in regenerative medicine including aging and neurodegenerative diseases, the exact molecular mechanism of β-HB has not been fully explored. In addition, exogenous β-HB contributed to stem cell homeostasis and intestinal stem cell function via Notch signaling activation for tissue regeneration [75]. Therefore, β-HB could be regarded as a potential regenerative mediator as well as alleviate age-related diseases (Figure 5).

### 3.2. Age-Related Diseases

#### 3.2.1. Cancers

Aging is a critical risk factor in the development of cancer, which is a primary cause of human death [76]. Plenty of energy is needed for cancer cells to enhance their proliferation rate. Cancer cells generate energy mainly by aerobic glycolysis, which is known as the Warburg effect [77]. This is regarded as an adaptive response of permitting carbons to be delivered via anabolic pathways and getting rid of the mitochondrial dysfunction induced by increased ROS levels. Thus, inhibiting glucose availability in cancer cells is considered an effective option. Recent animal studies showed that KD provides a therapeutic approach to tumor cells through selective metabolic oxidative stress, while simultaneously inhibiting primary tumorigenesis and systemic metastasis [78,79,80,81]. Thus, the published data indicated that KD may not significantly affect advanced and terminal cancers progression, yet it is safely used as an adjuvant therapy with other verified anticancer therapies and has the potential to improve the quality of cancer patients’ life [82,83]. It is suggested that this therapy should be further investigated to figure out the exact connections between β-HB and various types of cancer.

#### 3.2.2. Neurological Disorders

There are multifactorial complex underlying pathophysiological mechanisms in neurodegenerative diseases. Ketogenic nutrition or exogenous β-HB is reported to have the capability to weaken the progression of pathological conditions such as Alzheimer’s disease (AD) and PD. Neurodegenerative diseases have the characteristic of defects in mitochondrial metabolism, including attenuated ATP generation, increased ROS production, complex IV dysfunction in AD, and complex I dysfunction in PD [84,85,86]. β-HB may promote mitochondrial metabolism via multiple mechanisms such as inducing mitochondrial turnover, reducing oxidative stress and impairment of mitochondria, as well as serving as an alternative energy substrate [87,88]. Moreover, β-HB supplementation may resist neuroinflammation through inhibiting pathologic microglial activation and regulating NOD-like receptor family pyrin domain containing 3 (NLRP3) inflammasome pathways [89,90]. Finally, β-HB could prevent the toxicity of neurotoxins of AD and PD [91]. It is obvious that not only by single intervention or lifestyle change such as drug, diet, exercise, and sleep, but combining ketone supplements along with these changes, may improve the neurological debilitating diseases.

#### 3.2.3. Cardiovascular Diseases

Both morbidity and mortality of cardiovascular diseases (CVDs) increase with age, which is the central risk factor of many forms of major diseases. Different modalities of CR are regarded as effective interventions of extending lifespan on the aging process [92]. CR has been proven to produce positive defense responses in a variety of stress states, of which cardiovascular protective signaling is the core response, including mammalian target of rapamycin (mTOR), AMP-activated protein kinase (AMPK), sirtuins, and endothelial nitric oxide synthase signaling pathway [93]. KD is one kind of CR, which is closely associated with anti-aging impacts, during which high levels of β-HB are produced. Studies in mice have already demonstrated the role of β-HB in terms of extending healthspan, regulating HDACs, the enzymatic activity of epigenetic regulators, and thereby activating the expression of age-related genes [94]. β-HB therefore may be a powerful factor in attenuating cardiovascular aging which requires future research.

#### 3.2.4. Muscle Dysfunction

Age-related muscle dysfunction is characterized by progressive sarcopenia and atrophy of skeletal muscles, which is related to frailty, muscle weakness, and disability in the elderly [95]. The mechanisms underlying this process including some dominant features, such as inhibited anabolic pathways, enhanced catabolic pathways, loss of muscle fiber, and muscle denervation in both older human and rodents’ studies [96]. Recently, the anticatabolic, anabolic, and regenerative potential of β-HB has been demonstrated by therapeutic ketosis in human skeletal muscle atrophy under an inflammatory microenvironment. [97,98]. β-HB has also been proved to potentially slow muscle loss with myopathies by maintaining mitochondrial respiration and morphology within muscle tissue [99]. Cell and animal studies have particularly suggested that β-HB induced by exercise or provided by exogenous supplementation might improve skeletal muscle and cognitive function [100,101]. HDACs have been reported to be associated with age-related muscle dysfunction, playing crucial roles in regulating metabolic processes in skeletal muscle [102]. β-HB, which is one kind of HDAC inhibitor, could be a promising target to treat sarcopenia clinically.

#### 3.2.5. Inflammation

It is generally acknowledged that inflammatory mechanisms are closely associated with age-related diseases, such as arthritis, atherosclerosis, metabolic syndrome, fatty liver disease, type 2 diabetes, frailty, cachexia, cancer, and neurodegenerative disorders [103,104,105,106]. Long-term stimulation of immune response may lead to progressive degeneration of both innate and adaptive immune systems, causing decreased immunological competence, which is considered as “immunosenescence”. Chronic low-grade inflammation occurs throughout this immunological process which is named “inflammaging” [107]. β-HB is now considered a modulator of inflammation and immune cell function, yet various underlying mechanisms remain controversial [17]. β-HB administration could promote anti-inflammatory actions, involving the regulation of NLRP3 inflammasome in neutrophils and macrophages and the declining production of inflammatory molecules. [108]. Inflammation is an important component of many age-related diseases; thus, an anti-inflammatory response can be attributed to β-HB protection as recently reported [109].

#### 3.2.6. Metabolic Syndrome

Due to multiple physiological mechanisms, aging is becoming one of the most critical risk factors for the occurrence and progression of metabolic syndrome, which is an age-related disease consisting of obesity, glucose intolerance, insulin resistance, dyslipidemia, and hypertension [110]. Endoplasmic reticulum (ER) stress is disturbed homeostasis leading to an impaired protein synthesis process such as the accumulation of unfolded and misfolded proteins, which is especially associated with the onset of metabolic disorders such as diabetes mellitus and disorderly hepatic lipid metabolism [111]. Nonalcoholic fatty liver disease has been regarded as a novel component of metabolic syndrome, with attenuated hepatic ketogenesis, insulin sensitivity, and abnormal fat accumulation [112,113]. NK has been found to improve metabolic and inflammatory markers, as well as lower insulin levels and promote β-HB production particularly during KD [114]. Of note, β-HB administration has already been reported to inhibit inflammasome formation, lipid accumulation, and oxidative stress by binding to specific HCARs, and inhibiting HDACs, FFARs, and NLRP3, to suppress ER stress, suggesting the beneficial effects of β-HB supplementation on liver steatosis and restoration of liver functions in aging progression [115,116,117].

## 4. Therapeutic Application of β-HB in Aging

### 4.1. Endogenous Ketosis

Endogenous ketosis can be achieved by using various durations of fasting, CR, or CHO restriction and particularly by a KD [118] (Figure 5).

#### 4.1.1. Intermittent Fasting and Caloric Restriction

Energy-restricted metabolic states such as intermittent fasting (IF) or CR, have obvious characteristics of increased ketosis and could extend lifespan in animals [119]. Ketogenesis is believed to increase with prolonged starvation. IF is relatively easy to practice for a long time, with alternative periods of feeding or fasting which can last 24 h from one to four days per week, such as alternate-day fasting, whole-day fasting (periodic fasting), and time-restricted feeding [120,121]. IF has been proven to be safe in monitored patients and the hormonal level of this state is typically marked by a low-insulin, high-glucagon, and increased plasma fatty acids and cortisol environment which heavily promote lipolysis [122,123]. It is currently gaining more popularity and is being considered as a potential non-pharmacological way to promote healthy aging [124]. CR is achieved by reducing energy intake of about 25–30% without lacking essential nutrients, which has also been observed to improve age-related mortality and morbidity, delay aging progression, and result in healthspan in invertebrate and vertebrate species [125,126]. Circulating β-HB level is elevated as a beneficial metabolite and mediator during these two states, which are widely accepted as anti-aging interventions [127]. However, before these regimens, it is hard to maintain long-term ketosis. The mechanism of β-HB as a potential CR mimic to slow aging has yet to be explored further.

#### 4.1.2. Ketogenic Diets

A large amount of data on the effects of β-HB metabolism comes from studies on KD, especially in rodents. KD is not an energy-limiting state, yet the related phenotype replicates some of the biochemical properties of IF and CR which are strongly associated with longevity. KD is composed of high-fat, adequate-protein, and a very low level of CHO (typically about 88%, 10%, and 2%). The KD promotes endogenous ketogenesis without fasting [128]. In terms of lifespan extension, β-HB has been proposed to promote longevity in worms via two different anti-aging pathways, which are inhibiting HDACs, leading to increased DAF-16/FOXO activity, as well as involving the mitochondrial metabolism of β-HB, and activating the SKN-1/Nrf2 antioxidant response pathway [73]. KD has also been demonstrated to improve the longevity and survival of mice, together with increased protein acetylation and decreased activation of tissue-specific mTOR complex 1 [71]. KD has been mechanistically investigated to improve neuroprotection and mitochondrial metabolism, activate autophagy, enhance antioxidative and anti-inflammatory capability, and inhibit insulin/insulin-like growth factor signaling, which contributes to the anti-aging process [129]. Although KD has already been clinically used as a therapy, which is easier to sustain than CR, it is in some ways difficult to rigorously follow and requires specific medical guidance and strong motivation [130].

### 4.2. Exogenous Ketosis and Supplementation of β-HB

It has been demonstrated that not only KD, but also exogenous ketone supplements (EKSs) can increase and maintain blood KB level, especially β-HB, so as to promote anti-aging effects [131]. β-HB supplements are now being commercially marketed as an alternative to KD. The supplements are commonly present in either a powder form of ketone salts (KS) or a liquid form of ketone ester (KE). In addition, medium-chain triglycerides (MCTs) or their combination with MCTs oil are also usually used to induce and sustain NK to improve ketotic response [132,133]. The production of β-HB from these supplements would not be affected by CHO, thus administration of EKSs may be practical and alternative when maintaining a normal diet to achieve therapeutic ketogenesis (Figure 5) [134].

#### 4.2.1. Ketone Salts

Oral administration of isolated β-HB would be the most direct method of exogenously inducing NK. However, KB in its free acid form can be expensive, unstable, and ineffective at producing sustained ketosis. Thus, ketone acids buffering with sodium, potassium, calcium, or other electrolytes have been explored to enhance efficacy, inhibit overload of any single mineral, and these compounds are commercially available. It is reported that co-ingestion with MCTs may improve the efficiency of increasing β-HB relatively, at least in rats [135]. However, a few undesirable adverse effects exist while consuming large doses of KS, which usually results in gastrointestinal distress, and inappropriate cation overload or acidosis [136].

#### 4.2.2. Ketone Esters

Several existing synthetic KEs prove to be the most effective agents to induce immediate, sustained, and dose-dependent elevation in serum ketones concentration, which provides an alternative way to increase β-HB and is well-tolerated in rodents and humans [137,138]. Ester bonds hold KEs together and are cleaved by gastric esterases to release KB in their free acid form from the backbone molecule, which is often a ketogenic precursor molecule R,S-1,3-butanediol (BD). Two prominent KEs in the recent research studies are (R)-3-hydroxybutyl (R)-3-hydroxybutyrate ketone monoester (KME) and R,S-1,3-butanediol acetoacetate ketone diester (KDE), the former appears safer and superior at appropriate doses in healthy adults, whatever acutely or daily sustained up to 28 days [139,140,141,142]. NK produced by KEs is therefore achieved without prolonged fasting or KD yet it is currently the most potent method of EKSs.

#### 4.2.3. Medium Chain Triglycerides

MCTs have a much greater ketogenic potential than long-chain fatty acids since they are rapidly absorbed, energy-dense, water-miscible, and tasteless. 6 to 12 carbons of fatty acids are contained in MCTs in length. MCTs can be hydrolyzed to medium-chain fatty acids by lipases in the gastrointestinal tract, and then rapidly metabolized to Ac-CoA, finally to KB in the liver [143]. MCTs are consequently regarded as ketogenic fats due to their ability of ketogenesis without the restriction of dietary CHO intake [144]. Unfortunately, high MCTs consumption are often not well adopted because of their gastrointestinal side effects, including diarrhea, dyspepsia, and flatulence, which could be alleviated through a progressive 1 or 2-week period [145]. In addition, the generation of β-HB by supplementation of MCTs is at a low level in the blood [146].

#### 4.2.4. R, S-1,3-Butanediol

BD is an organic butyl alcohol approved by the Food and Drug Administration, which is metabolized to produce two isoforms of β-HB, D- and L-β-HB or R- and S-β-HB via hepatic conversion, even though it is not a fatty acid or MCT [147]. Oral administration of BD could achieve ketosis and approach a KD state in dogs [148]. It was demonstrated that a dose-dependent elevation of KB in a ratio of 6:1 of R-β-HB to AcAc could be produced by BD in rodents [149]. BD is often utilized as a backbone in the synthesis of KE. Gut or tissue esterases could easily break the ester bond and release KB and BD without the involvement of salts or acid [150]. A variety of preclinical toxicology studies have found that BD is safe and tolerable [151].

#### 4.2.5. β-HB Enantiomers

β-HB is a chiral molecule, with two enantiomers, R/D and S/L, which is an important characteristic in terms of its signaling activities as well as possible therapeutic applications [152]. Currently, a racemic mixture enantiomer of β-HB is the most commercially available ingestible EKSs apart from KS and KME, as its synthesis is more affordable than pure enantiomers. The chiral specificity is introduced by BDH1, determining that only R-β-HB is the normal product of human metabolism and could be readily catabolized into ATP and Ac-CoA [153]. IF, CR, KD, exercise, or any other situation which leads to endogenous β-HB would produce only R-β-HB. It is reported that ingestion of the same amount of racemic EKSs may produce higher and more sustained S-β-HB in blood circulation due to its slower metabolization compared to R-β-HB [154,155]. Despite divergent metabolic effects, these two enantiomers have similar molecular interactions as well as intracellular signal transduction cascades, which remains a hotly debated topic.

### 4.3. Comparisons between Endogenous and Exogenous Ketosis Induced by NK

In essence, the metabolic conditions of chronic endogenous dietary ketosis are in obvious contrast to the rapid exogenous ketosis delivered by ketone bodies. The following are the differentiations. Firstly, KD elevates blood β-HB level to a range of 0.5 mM to 3.0 mM, whereas EKSs approximately elevate to 0.3 mM to 1.0 mM [156]. Secondly, KD requires a couple of days to achieve sustained NK state, while EKSs elevate β-HB concentration acutely. Thirdly, KD need to follow strict CHO intake while EKSs require no direct CHO restriction, which determines EKSs has higher compliance, especially in short-term supplementation [142]. Fourthly, circulating glucose concentrations are therefore divergent because of the diverse CHO requirement. Fifthly, their similar anabolic and anticatabolic effects have as well been demonstrated [157,158]. Finally, KD and EKSs have been reported to reduce the substrate utilization of CHO during exercise, yet under distinct metabolic states [159,160]. NK is defined as a metabolic state, exerting physiological changes at both systemic and cellular level wherein β-HB concentration is over 0.5 mM regardless if induced by endogenous or exogenous ketones. These effects can be similar or different and can be universal or tissue-specific. NK is believed to be a potential state for performance-enhancing or therapeutic benefits under endogenous or exogenous ketosis, which need further evaluation.

## 5. Future Perspectives

Future studies are necessary to further elucidate the following practical issues: activating endogenous NK in a normal dietary context to sustain a steady state of metabolism; improving the targeted delivery of β-HB prodrugs or precursors to avoid excessive salt load or acidosis; bringing β-HB to the sites of action by using existing endogenous transporters and metabolite gradients to explore specific downstream signaling events; confirming whether the synthetic KE compounds need been strictly pure due to its enormous financial burden for the majority of patients and health systems; investigating if S-β-HB has a better pharmacokinetic than R-β-HB, which might help reduce cost and discover another signaling function; establishing methodologies to quantify β-HB flux rates and differentiate these two enantiomers in terms of concentrations and impacts. Besides, the recommended dose and timing of β-HB supplements, the short half-life and bitter taste of KME as well as the interaction with other substrates in various nutritional surroundings are also needed to be specified. β-HB is emerging as vitally important regulators of metabolic health and longevity, alleviating aging phenotypes via multiple and yet unknown molecular mechanisms. By modulating lipolysis, energy expenditure, metabolic rate, insulin resistance, autophagy, feeding behavior, as well as exercise performance, β-HB might serve as a signaling biomolecule to affect cellular function and human healthspan. The evaluation of β-HB may be a crucial approach for the treatment of the aging population.

## 6. Conclusions

In conclusion, β-HB is the most abundant KB and plays a vital role as an energetic metabolite, a signaling molecule, as well as an epigenetic regulator, which could be used as a therapeutic agent in a range of cancers, neurodegeneration, traumatic brain disorders, cardiac diseases, muscle dysfunction, metabolic syndrome, and inflammation. Endogenous ketosis and an exogenous supplement may be promising strategies for numerous diseases. Further research is needed to investigate whether ketotherapeutics can promote healthy aging, and to figure out the specific relationship and underlying mechanisms between β-HB and the aging process, which may offer a novel way in delaying the onset and development of age-associated dysfunctions.

## Figures and Tables

**Figure 1 nutrients-13-03420-f001:**
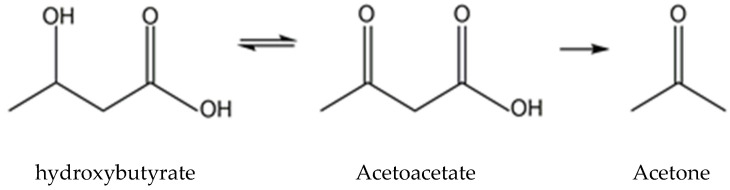
Structure of ketone bodies.

**Figure 2 nutrients-13-03420-f002:**
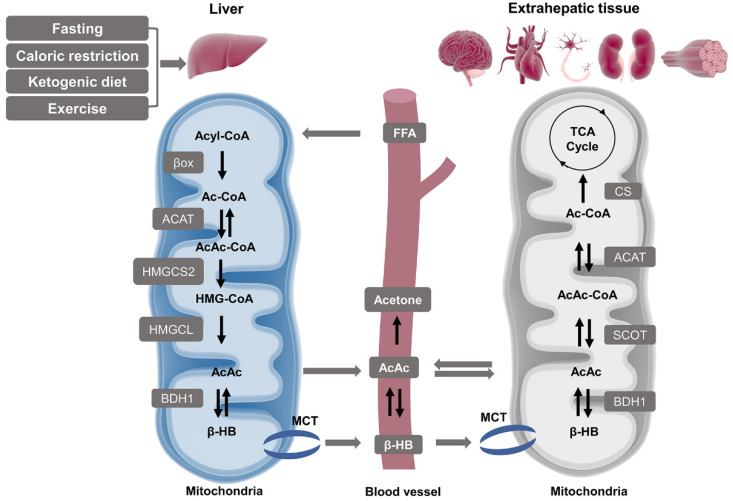
Pathways of ketogenesis in liver and ketolysis in extrahepatic tissues. Hepatic mitochondria act as the primary site for the synthesis of blood ketone bodies by using fatty acids-derived Ac-CoA which is generated by β oxidation. The following process requires four enzymes: ACAT, HMGCS2, HMGCL, and BDH1, and the intermediate product covers AcAc-CoA, HMG-CoA, and AcAc. β-HB is finally produced and released to the bloodstream and uptaken by extrahepatic tissues such as the brain, heart, neurons, kidneys, and muscles through MCT. Both β-HB and AcAc can be oxidized and converted to Ac-CoA and produce ATP via the TCA cycle as an alternative energy source. Abbreviations: AcAc, acetoacetate; AcAc-CoA, acetoacetyl CoA; ACAT, acetyl-CoA A acetyltransferase; Ac-CoA, acetyl CoA; BDH1, β-hydroxybutyrate dehydrogenase 1; β-HB, β-hydroxybutyrate; βox, β oxidation; CS, citrate synthase; FFA, free fatty acid; HMGCL, HMG-CoA lyase; HMGCS2, 3-hydroxymethylglutaryl-CoA synthase 2; HMG-CoA, 3-hydroxymethylglutaryl-CoA; MCT, monocarboxylate transporter; SCOT, succinyl-CoA:3-oxoacid CoA transferase; TCA, tricarboxylic acid.

**Figure 3 nutrients-13-03420-f003:**
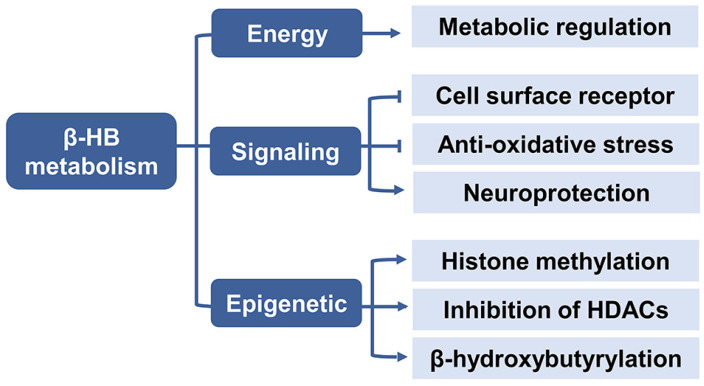
The metabolic role of β-HB. β-HB acts as an energy substrate to regulate metabolic response, and serves as a signaling molecule to inhibit lipolysis, oxidative stress and improve neuroprotection, it also plays a role in epigenetic regulation including histone methylation, inhibition of HDACs, and histone lysine β-hydroxybutyrylation. Abbreviations: β-HB, β-hydroxybutyrate; HDAC, histone deacetylases.

**Figure 4 nutrients-13-03420-f004:**
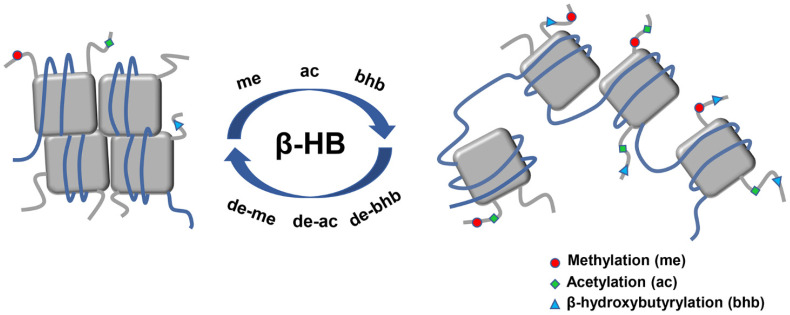
Main epigenetic alterations of histone posttranslational modifications induced by β-HB. Histone posttranslational modifications which relate to the epigenetic regulation of β-HB mainly consist of histone methylation, acetylation, as well as a new type of histone lysine bhb. β-HB administration alters the modification of chromatin conformation through the epigenetic pathway. Chromatin structure is represented by DNA (blue line) organized in nucleosomes formed by core histones (grey square). Abbreviations: ac, acetylation; bhb, β-hydroxybutyrylation; β-HB, β-hydroxybutyrate; me, methylation.

**Figure 5 nutrients-13-03420-f005:**
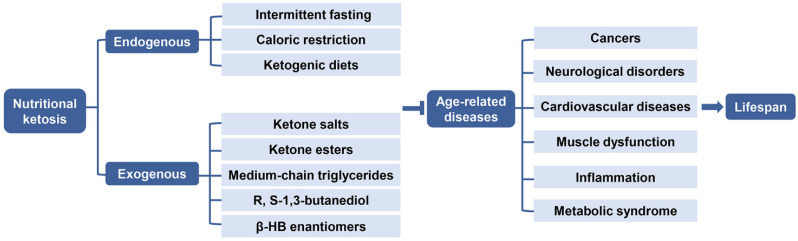
Role of endogenous and exogenous nutritional ketosis in age-related diseases. Nutritional ketosis can be achieved endogenously through intermittent fasting, caloric restriction, ketogenic diets or exogenously with oral supplementation via ketone salts, ketone esters, medium-chain triglycerides, R, S-1,3- butanediol, and β-HB enantiomers, resulting in a therapeutic effect in several medical conditions, thereby targeting the underlying mechanisms of age-related diseases, such as cancers, neurological disorders, cardiovascular diseases, muscle dysfunction, inflammation, and metabolic syndrome, which may extend the healthy life expectancy. Abbreviations: β-HB, β-hydroxybutyrate.

## Data Availability

Not applicable.

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
