# Peer review of "?-hydroxybutyrate as an Anti-Aging Metabolite"

_nutrients, 2021, doi:10.3390/nu13103420_

Round 1

Reviewer 1 Report

The present work covers an interesting topic but I think that has more than one thing to be reviewed:

line 34 ref 2 and 3 are not pertinent and it doesn't justify the previous sentences, especially referring to the ketonemia

line 37 ref 4 and 5 are on mice and doesn't refer to a general mechanism

line 101 heart doesn't need ketones because it's able to use regularly fatty acid

line 116 it's just a hypothesis, as stated in the first citation

IF section should be clarified because it depends on how long lasts fasting time and it depends on how is set the caloric intake on eating time

The ketogenic diet section should be rewritten, most of the side effects are reported in the ketogenic diet operated in the wrong manner and/or too small caloric amount, the ketogenic diet could be even hypercaloric and applied in the sport practice too...

PEK section is probably unnecessary because the ketosis will last for a too-short time

All exogenous ketones present various problems:

Intestinal discomfort, even if used in the form of salts

The risk of having ketoacidosis, as if ketosis is induced (fasting or lack of carbohydrates) the blood sugar will certainly be medium-low, but in the case of exogenous intake it could be high, thus leading to ketoacidosis, so the glycemia

The cost is very high, so it is difficult to think of supplementation for a long time

At the moment the affective action of ketones alone is still to be clarified, most of the work is a hypothesis, the only consolidated by literature is the ketogenic diet, which probably works on the overall and not only thanks to ketones presence

Reviewer 2 Report

The authors present a review on the role of BHB in anti-aging. 

The review is well written and easy to follow. 

Major request:

1) It is important to add a paragraph on the difference between "functional ketosis and chronic ketosis". How are they obtained and which are the effects on cell metabolism? 

Round 2

Reviewer 2 Report

I thank the reviewer for adding the paragraph "Comparisons between endogenous and exogenous ketosis induced by NK". 

Author Response

Dear editors, Dear reviewers,

Thank you very much for giving us an opportunity to revise the manuscript. We thank the editors and reviewers for the time and effort. Their valuable suggestions have enabled us to improve our work. The manuscript has been edited by an English-speaking native. Many grammatical errors have been modified according to the helpful advice.  

Here, we attached revised manuscript with all the changes highlighted by using the track changes mode in MS Word. Appended to this letter is our point-by-point correction to the comments raised by the academic editor. The comments are reproduced and our responses are given directly afterward in a different color.

We look forward to hearing from you regarding our submission. We would be glad to respond to any further questions and comments that you may have.

Sincerely,

Weihua Xiao, MD, PhD

Shanghai Frontiers Science Research Base of Exercise and Metabolic Health,

Shanghai University of Sport,

No. 188, Hengren Road, Wujiaochang Town, Yangpu District, Shanghai 200438, China

E-mail: xiao_weihua@163.com

Tel: +86 021-65507367  

Manuscript ID: nutrients-1353660

Type of manuscript: Review

Title: β-hydroxybutyrate as an anti-aging metabolite

Authors: Lian Wang, Peijie Chen *, Weihua Xiao *

Received: 8 August 2021

E-mails: wanglian9369@hotmail.com, chenpeijie@sus.edu.cn, xiao_weihua@163.com

Submitted to section: Nutrition and Public Health,

https://www.mdpi.com/journal/nutrients/sections/Nutrition_Public_Health

(1)  Abstract lines 10-18 second sentence extensive length and unclear as written.

Page 1 line 12-15

(2)  Page 2 line 50 “supplementation of D-BHB to C. elegan” unclear

Page 2 line 55-58

(3)  Lines 46-53 delete through end of sentence … its real biological significance. Start with “This review will …

Page 2 line 58

(4)  Line 65 add s to fatty acid

Page 2 line 70

(5)   Line 65 delete be into and insert enter the

Page 2 line 70

(6)   Line 66 add s to fatty acid

Page 2 line 71

(7)   Line 67 delete “and … how” in the sentence. Reword next sentence

Page 2 line 71-72

(8)   Lines 68-70 sentence is not clear as written

Page 2 line 72-74

(9) Page 3 line 93 period after word oxidation. Lines 92-94 unclear phrase, “which for , metabolic response to …. Not clear what “which for” refers to

Page 3 line 107-111

(10) Line 102 unclear “CHO content” do you mean inadequate dietary CHO intake?

Page 3 line 119

(11) Line 103 delete extreme exercise or define it

Page 3 line 119

(12) Lines 104-105 sentences unclear as written. What is the difference between “fats and lipids”, what is meant by energy requirement. Do not think this is the correct word choice

Page 3 line 120

(13) Lines 106-108 grammar run on sentence

Page 3 line 121, Page 4 line 134-135

(14) Line 113 move word daily after word produced

Page 4 line 141

(15) Line 114 add “the” after of and add s to requirement

Page 4 line 141-142

(16) Line 117 change or to and

Page 4 line 144

(17) Line 118 change or to and

Page 4 line 145

(18) Page 5 line 146 delete word have

Page 5 line 180

(19) Line 159 unclear meaning of “alleviate” glucose uptake. Word choice?

Page 5 line 193

(20) Line 173 delete “entail”

Page 5 line 207

(21) Line 180 change proved to shown

Page 5 line 214

(22) Page 7 Line 222 change improve to promote

Page 7 line 261

(23) Line 223 add word is after it

Page 7 line 262

(24) Line 225 What is C. elegans?

It has been explained in Page 2 line 55-58.

(25) Lines 227-230 “A variety ….” Sentence needs rewording. Unclear as written

Page 7 line 266-268

(26) Line 243 change progression to development

Page 7 line 281

(27) Lines 249-251 “It has been …. animal studies “ sentence unclear as written

Page 7 line 287-289

(28) Line 254 delete promote insert improve

Page 7 line 292

(29) Page 8 Lines 276-277 phrase “which seem to impinge …” unclear what the phrase is referring to.

Page 8 line 328

(30) Lines 277-280 “CR …state” sentence unclear as written. Does not relate to the second part of the sentence which is also unclear as written. Section needs to be rewritten and focused

Page 8 line 329-332

(31) Lines 291-293. Grammar issues

Page 8 line 343-346

(32) Line 298 change supported to reported

Page 8 line 350

(33) Delete last sentence of the paragraph or provide direct references

Page 8 line 353

(34) Lines 312-315. Grammar

Page 8 line 363-364, Page 9 line 379-380

(35) Page 9 line 329 delete clause  “, which has been …. Decades.

Page 9 line 395

(36) Line 339 add word and after restriction and add a before KD

Page 9 line 403

(37) Line 348 delete , after acids and add “and” after glucagon

Page 9 line 411-412

(38) Line 349 delete s off promotes

Page 9 line 413

(39) Page 10 lines 350-351 CR is … insert word achieved and delete “an intervention”.

Page 9 line 415

(40) Line 356 Insert the word “before” these, add a , after regimens, add it is hard and delete are. Add . after ketosis and delete and

Page 9 line 420

(41) Line 357 start new sentence The mechanism

Page 9 line 420

(42) Line 360 take s off comes

Page 9 line 423

(43) Lines 360-362  grammar issues

Page 9 line 423-426

(44) Line 364 add . after ). Start sentence The KD promotes and eliminate “which… to”

Page 9 line 427

(45) Line 375 add to s to way

Page 10 line 462

(46) Page 11 line 399 cannot find definitions of EKSs, and define and give amounts

Page 10 line 465-466

(47) Line 407 Add . after KD.  The supplements are commonly present …diet,

Page 10 line 468

(48) Line 408 define KE and line 411 define KE

Page 10 line 469

(49) Line 412 practical alternative

Page 10 line 472-473
